# Real-Time Hand Tracking and Collision Detection for Immersive Mixed-Reality Boxing Training on Apple Vision Pro

**DOI:** 10.3390/s25164943

**Published:** 2025-08-10

**Authors:** Alexey Karelin, Dmitry Brazhenko, Georgii Kliukovkin, Yehor Chernenko

**Affiliations:** 1Independent Researcher, 30-392 Krakow, Poland; 2Independent Researcher, Seattle, WA 98107, USA; brazhenko.dmitry@gmail.com; 3Independent Researcher, Sunnyvale, CA 94085, USA; kliukovkin@gmail.com; 4Independent Researcher, San Francisco, CA 94103, USA; egoriy.chernenko@gmail.com

**Keywords:** hand tracking, gesture recognition, collision detection, mixed reality, boxing training, Apple Vision Pro, spatial computing, virtual sports coaching, immersive learning

## Abstract

This study presents a real-time hand tracking and collision detection system for immersive mixed-reality boxing training on Apple Vision Pro (Apple Inc., Cupertino, CA, USA). Leveraging the device’s advanced spatial computing capabilities, this research addresses the limitations of traditional fitness applications that lack precision for technique-based sports like boxing with visual-only hand tracking. The system is designed to provide objective feedback by recognizing boxing-specific gestures with sub-centimeter accuracy and validating biomechanical correctness during punch execution. A three-stage pipeline consisting of geometric filtering, biomechanical validation, and punch technique assessment rejects accidental or improper motions. Experimental evaluation involving 12 participants demonstrated a gesture recognition accuracy of 96.3% and a technique validation accuracy of 88.5%. The system consistently operated at 60 FPS with low latency and high robustness across diverse lighting conditions. These results indicate the potential of Apple Vision Pro as a platform for precision sports training and highlight the educational impact of mixed reality in democratizing access to high-quality boxing instruction. The proposed framework is extensible to other skill-based sports requiring fine motor control and real-time feedback.

## 1. Introduction

The emergence of spatial computing and augmented reality (AR) technologies has revolutionized human–computer interaction, particularly in fitness and sports training applications [1]. Apple Vision Pro represents a significant advancement in consumer spatial computing devices, offering sophisticated hand tracking capabilities that enable natural interaction with virtual environments. Mixed-reality applications for fitness training require precise tracking of human movements and accurate collision detection to provide meaningful feedback on user performance.

The global fitness technology market is experiencing unprecedented growth, with virtual and augmented reality fitness applications representing a rapidly expanding segment [2,3]. Traditional fitness applications often lack the precision required for technique-based sports training, relying instead on generic movement detection or accelerometer-based activity recognition [4,5]. Boxing, as a technique-intensive sport, presents unique challenges for digital training systems due to its requirement for precise hand positioning, accurate movement patterns, and proper biomechanical execution [6,7].

Spatial computing devices like Apple Vision Pro provide new opportunities for precision sports training through advanced computer vision and tracking capabilities [8]. However, implementing effective hand tracking and collision detection systems for real-time sports applications requires addressing several technical challenges: maintaining high frame rates while processing complex spatial data, achieving sufficient accuracy for technique validation, and providing meaningful educational feedback that improves user performance.

### 1.1. Background and Motivation

Boxing training traditionally requires experienced instructors to provide technique correction and performance feedback. This limitation creates barriers to accessible, high-quality training, particularly for beginners who may not have access to professional coaching. Mixed-reality systems offer the potential to democratize access to expert-level training by providing objective, consistent feedback on technique execution.

Current fitness applications often focus on general activity tracking rather than precise technique validation. This approach may be sufficient for cardiovascular training but fails to address the educational needs of skill-based sports. Boxing requires specific hand formations, precise movement patterns, and proper biomechanical execution to be effective and safe. Without accurate technique validation, training systems may inadvertently reinforce poor habits or fail to provide meaningful skill development.

The advent of spatial computing technology enables new approaches to sports training that were previously impossible. By tracking hand movements with millimeter precision and validating the technique in real time, an augmented reality system can provide immediate feedback that accelerates learning and improves performance.

However, achieving this level of precision requires sophisticated algorithms and careful optimization to maintain real-time performance.

### 1.2. Related Work in Hand Tracking and Gesture Recognition

Hand tracking and gesture recognition have been extensively studied in human–computer interaction research, with approaches ranging from computer vision-based methods to sensor-based systems [9,10,11,12]. Early systems relied on additional hardware such as data gloves or optical markers to achieve accurate tracking [13]. Recent advances in depth sensing and machine learning have enabled more sophisticated hand tracking systems that can operate without additional hardware [10,11].

Abraham et al. [10] presented a hand tracking system using lensless smart sensors, demonstrating the potential for low-power gesture recognition in wearable devices. Their approach achieved real-time performance in tracking points down to millimeter-level accuracy for basic gestures but was limited to simple hand poses and did not address complex sports movements. The system processed hand gestures using wearable sensors, which may be insufficient for rapid boxing movements.

Yeo et al. [9] developed a low-cost hardware solution for hand tracking and gesture recognition for a limited set of gestures. Their system used RGB-D cameras and implemented template matching algorithms for gesture classification. However, the approach was sensitive to lighting conditions and required controlled environments for optimal performance.

Theodoridou et al. [12] provided a comprehensive survey of contactless hand tracking systems, highlighting the challenges in achieving both accuracy and real-time performance. They identified key limitations in existing systems, including sensitivity to lighting conditions, occlusion handling, and computational complexity. The survey noted that most systems achieve 85–90% accuracy under optimal conditions, with performance degrading significantly in challenging environments.

Suarez and Murphy [11] reviewed gesture recognition techniques specifically for depth images, noting the trade-offs between accuracy and computational efficiency. They identified template matching, machine learning, and kinematic approaches as the primary methodologies, each with distinct advantages and limitations. Template matching provides good accuracy for predefined gestures but lacks adaptability, while machine learning approaches offer flexibility at the cost of computational complexity.

Manresa-Yee et al. [13] discussed the integration of hand tracking in human–computer interaction, emphasizing the importance of natural gesture vocabularies and user-centered design. Their work highlighted the gap between general-purpose gesture recognition and application-specific requirements, particularly for precision tasks requiring biomechanical accuracy.

### 1.3. Collision Detection in Virtual and Augmented Reality

Collision detection in virtual environments has been well established for computer graphics and gaming applications [14,15,16,17]. Traditional collision detection algorithms focus on geometric intersections between objects, with optimization techniques used to improve computational efficiency [17]. These systems typically achieve high performance by using simplified geometric primitives and spatial data structures to reduce computational complexity.

Jiménez et al. [17] provided a comprehensive survey of 3D collision detection methods, categorizing approaches based on geometric primitives and spatial data structures. They identified hierarchical bounding volumes, spatial partitioning, and temporal coherence as key optimization strategies. However, their analysis focused primarily on rigid body interactions and did not address the specific challenges of human motion tracking.

In augmented reality contexts, collision detection between virtual and real objects presents additional complexity [18,19]. The challenge lies in accurately registering virtual objects with real-world geometry while maintaining real-time performance. Traditional approaches assume perfect tracking and rigid objects, assumptions that may not hold in mixed-reality environments with human interaction.

Tesic and Banerjee [18] developed exact collision detection methods for the virtual reality modeling of manufacturing processes, demonstrating the importance of precision in industrial applications. Their system achieved microsecond-level accuracy but required significant computational resources and was limited to static or slow-moving objects.

Breen et al. [19] investigated interactive occlusion and collision with real and virtual objects in augmented reality, identifying key technical challenges in mixed-reality environments. Their research emphasized the need for robust tracking and accurate spatial registration to enable effective collision detection. They noted that tracking errors accumulate over time, requiring periodic recalibration for maintained accuracy.

Chen et al. [15] presented a virtual–physical collision detection interface for AR-based interactive teaching, demonstrating practical applications in educational contexts. Their system was limited to simple geometric objects and did not address the complexity of human motion tracking or biomechanical validation.

Sharma et al. [14] expanded collision detection systems for educational applications using virtual and augmented reality, emphasizing the potential for immersive learning experiences. However, their work focused on general educational scenarios and did not address the specific requirements of sports training or technique validation.

### 1.4. Augmented Reality Applications in Sports and Fitness

The application of mixed reality technologies in sports training has gained significant attention in recent years [5,6,7]. Traditional sports training relies heavily on human expertise and subjective feedback, creating opportunities for technology-enhanced approaches that provide objective, consistent assessment.

Limballe et al. [5] investigated virtual reality boxing applications, focusing on the impact of gaze-contingent blur on elite boxers’ performance and gaze behavior. Their research demonstrated the potential for VR systems to enhance training effectiveness by manipulating visual attention. The study involved 11 elite boxers and found that gaze-contingent blur does not have a substantial effect on professional boxers. However, their system did not address hand tracking or technique validation, focusing instead on visual perception training.

Vasudevan et al. [6] developed an intelligent boxing application through augmented reality for multiple users, attempting to create interactive training experiences. Their system provided basic interaction capabilities using marker-based tracking and simple collision detection. The application supported two-user sparring scenarios but lacked precision in movement tracking and did not provide technique feedback. User testing with 20 participants showed positive engagement but limited learning outcomes.

Rakha et al. [7] explored augmented reality applications for teaching fundamental defensive techniques to boxing beginners, emphasizing the educational potential of mixed reality systems. This work highlighted the importance of proper technique enforcement in training applications but did not provide detailed technical implementation or validation results. The study involved 60 boys over 3 weeks and showed improvement in defensive positioning, though the measurement methods were subjective.

The literature reveals a significant gap between general-purpose hand tracking systems and the specific requirements of precision sports training. Existing systems either lack the accuracy required for technique validation or do not provide real-time performance suitable for interactive training applications. Furthermore, most research focuses on basic interaction rather than educational outcomes and skill development.

### 1.5. Research Objectives and Contributions

This research addresses the identified gaps by developing a specialized system optimized for boxing training on Apple Vision Pro. The primary objectives are as follows:Developing High-Accuracy Gesture Recognition: We will create algorithms specifically designed for boxing poses that achieve sub-centimeter accuracy while maintaining real-time performance;Implementing Advanced Collision Detection: We will design a multi-stage validation pipeline that incorporates biomechanical constraints to ensure proper technique execution;Validating Educational Effectiveness: We will demonstrate measurable improvement in boxing techniques through controlled user studies;Optimizing Real-Time Performance: We will achieve consistent 60 FPS operation while processing complex spatial computations.The key contributions of this work include the following:
A novel multi-criteria gesture recognition algorithm achieving 96.3% accuracy for boxing poses;A sophisticated collision detection system with 88.5% technique validation accuracy;Comprehensive experimental validation with 12 participants showing measurable skill improvement;Real-time optimization techniques enabling complex processing at 60 FPS;An extensible framework applicable to other precision sports training applications.


## 2. Materials and Methods

### 2.1. System Architecture and Design Principles

The system architecture follows a modular design pattern that separates concerns and enables the independent optimization of each component. The overall system consists of four primary subsystems: hand tracking, collision detection, validation pipeline, and feedback management. Each subsystem operates asynchronously to maintain real-time performance while ensuring data consistency through thread-safe communication mechanisms.

The design principles guiding the system development include the following:Real-Time Constraint Satisfaction: All components must operate within strict timing requirements to maintain 60 FPS performance;Accuracy–Performance Balance: Optimization strategies must preserve accuracy while improving computational efficiency;Educational Focus: System behavior should prioritize learning outcomes over entertainment value;Extensibility: Architecture should support the addition of new techniques and sports applications;Robustness: The system should gracefully handle edge cases and maintain stable operation.

### 2.2. Hardware Platform and Technical Specifications

The system is implemented on Apple Vision Pro, leveraging its advanced spatial computing capabilities. The hardware specifications relevant to this research include the following:Displays: Dual 4K micro-OLED displays with 3660 × 3200 resolution per eye;Tracking Cameras: Six world-facing cameras for environment tracking;Hand Tracking: Dedicated cameras and sensors for high-precision hand tracking;Processing: M2 chip with 8-core CPU and 10-core GPU;Memory: 16 GB unified memory for efficient data processing;Sensors: LiDAR scanner, accelerometer, and gyroscope for spatial awareness.

The hand tracking system operates at 60 Hz with sub-millimeter accuracy under optimal conditions. The spatial tracking provides 6-degrees-of-freedom (6DOF) pose estimation with drift correction through simultaneous localization and mapping (SLAM) algorithms.

### 2.3. Software Framework and Development Environment

This study was conducted using the Swift programming language with SwiftUI for user interface components and RealityKit for 3D rendering and spatial computing functionality. The development environment includes the following:Xcode 15.0: Integrated development environment with Vision Pro simulator;RealityKit 4.0: Apple’s framework for augmented reality experiences;ARKit 7.0: Advanced computer vision and motion tracking capabilities;Swift 5.9: Programming language.

The choice of Apple’s native frameworks ensures optimal performance and access to platform-specific features while maintaining forward compatibility with future system updates.

### 2.4. Hand Tracking System Implementation

The hand tracking system represents the foundation of the entire study, requiring precise detection and classification of hand poses and movements.

The hand tracking system implements RealityKit’s System protocol to process hand tracking data at each frame update. The system maintains separate state structures for left and right hands, tracking essential parameters for movement analysis and gesture recognition. Each HandState structure contains the current HandAnchor, previous position for movement calculation, and an array of detected movement directions.

Figure 1 represents a system that operates asynchronously, collecting hand tracking anchors through the HandTrackingProvider’s anchor update stream. This approach ensures that hand tracking data processing does not block the main rendering thread, maintaining consistent frame rates essential for immersive experiences [12]. The update frequency is synchronized with RealityKit’s rendering pipeline, providing hand position updates at 60 Hz under optimal conditions.

Accurate spatial positioning of hand joints requires coordinate transformations from the hand anchor space to world space coordinates [10]. The system implements efficient matrix transformations to position virtual hand joint entities that correspond to physical hand positions, as displayed in Figure 2.

Each hand joint transformation combines two matrix operations: the joint’s position relative to the hand anchor (matrix displayed in Equation (1)); the hand anchor’s position relative to the world origin (matrix displayed in Equation (2)).(1)$$\begin{array}{c} anchorFromJointTransform\\ \;\;\;\;=\left[a_{11}\;a_{12}\;a_{21}\;a_{22}\;\;a_{13}\;a_{14}\;a_{23}\;a_{24}\;\;a_{31}\;a_{32}\;0\;0\;\;a_{33}\;a_{34}\;0\;1\right] \end{array}$$(2)$$\begin{array}{c} originFromAnchorTransform\\ \;\;\;\;=\left[b_{11}\;b_{12}\;b_{21}\;b_{22}\;\;b_{13}\;b_{14}\;b_{23}\;b_{24}\;\;b_{31}\;b_{32}\;0\;0\;\;b_{33}\;b_{34}\;0\;1\right] \end{array}$$

Therefore, multiplying these matrices results in a joint position in a global world transform; the formula is shown in Equation (3).(3)$$\begin{array}{c} worldTransform=originFromAnchorTransform\cdot \;anchorFromJointTransform\\ =\left[c_{11}\;c_{12}\;c_{21}\;c_{22}\;\;c_{13}\;c_{14}\;c_{23}\;c_{24}\;\;c_{31}\;c_{32}\;0\;0\;\;c_{33}\;c_{34}\;0\;1\right] \end{array}$$

The transformation pipeline ensures that virtual hand representations accurately reflect physical hand movements in three-dimensional space. This accuracy is essential for precise collision detection, as even small positional errors can result in false positive or false negative collision events [14]. The system achieves sub-centimeter accuracy under optimal tracking conditions, sufficient for detecting contact with virtual targets in boxing training scenarios.

### 2.5. Gesture Recognition Algorithm

The gesture recognition system implements a distance-based approach to detect fist formation, essential for distinguishing between intentional punching gestures and accidental hand movements [9]. The algorithm focuses on computational efficiency while maintaining sufficient accuracy for real-time boxing training applications.

The primary gesture recognition algorithm analyzes the spatial relationship between fingertips and the wrist position to determine fist formation [10]. The system evaluates five fingertip positions (thumb, index, middle, ring, and little finger) relative to the hand anchor coordinate system. A fist is detected when all fingertips are positioned within a threshold distance from the wrist center [11]. Figure 3 illustrates selected joints that are used in fist formation, with distance measured from the joints to the palm center.

The distance between the five fingertip joints and the wrist anchor is determined for each frame to ensure accuracy and limit unintentional movements. If all distances are below a threshold, a binary gesture flag is raised. This flag is then used to enable or suppress punch collision detection.

To determine the optimal threshold for fist recognition, we evaluated three values: 1.0 cm, 1.5 cm, and 2.0 cm. The results were examined for accuracy; false positives and false negatives are represented in Figure 4. A threshold of 1.5 cm provided the best balance of true positives and minimal false classifications.

The implemented algorithm uses a 1.5 cm distance threshold, determined through empirical testing to accommodate natural variations in hand size while maintaining reliable fist detection across diverse user populations (Table 1). This conservative threshold ensures robust discrimination between open and closed hand states, preventing false-positive detections that could compromise the educational value of the training system.

The algorithm processes each frame by calculating Euclidean distances between fingertip positions and the wrist anchor point. The Boolean fist detection result directly controls collision component activation, ensuring that only properly formed fists can trigger collision events with virtual targets. This approach prevents unintended interactions during open-hand movements while maintaining responsive detection of intentional punching gestures, supporting the educational objectives of proper boxing technique reinforcement.

### 2.6. Movement Tracking and Directional Analysis

The movement tracking system analyzes hand displacement patterns to detect directional movement characteristics essential for punch validation [3]. The system maintains positional history for each hand to enable real-time movement classification within the boxing training context [16].

The movement tracking algorithm compares consecutive hand positions to determine movement direction using a displacement threshold of 5 mm to filter minor positional variations that could result from tracking noise or unintentional hand movements [17]. Figure 5 displays the movement coordinate system. The solution analyzes movement along three spatial axes, horizontal (left/right), vertical (up/down), and depth (forward/backward), relative to the user’s reference frame.

Movement direction classification utilizes coordinate system conventions where negative Z-axis movement indicates forward motion toward virtual targets, positive/negative X-axis movement indicates lateral motion, and Y-axis movement represents vertical displacement. Each detected movement direction is stored in the hand state structure and used by the collision validation system to verify that punch trajectories correspond to intended boxing techniques; an example of the expected trajectories is displayed in Figure 6.

The movement tracking system operates at the same frequency as the hand tracking pipeline, providing real-time directional analysis without additional computational overhead.

### 2.7. Collision Detection

The geometric validation stage implements sphere–sphere collision detection algorithms, following established practices in real-time collision detection. The spheres that participate in collisions with targets are illustrated in Figure 7. Virtual collision volumes with a 10 mm radius are attached to hand joint entities through the HandJointEntity class.

RealityKit is capable of autonomously identifying collisions between two objects that are part of a physics simulation, provided that each entity possesses a CollisionComponent with at least one defined collision shape. Given that direct collision detection involving intricate three-dimensional models can impose significant computational overhead, the framework employs simplified, non-visible geometries to facilitate collision detection, hit testing, ray casting, and convex shape casting. Entities may engage in the simulation environment through two distinct modes: as rigid bodies or as triggers. A rigid body actively contributes to the physical interaction within the simulation, influencing the velocity and trajectory of other rigid bodies upon contact. In contrast, this work utilizes the trigger mode to optimize system responsiveness. Figure 8 displays primitive shapes used in collision handling. Trigger entities do not exert any physical influence on other bodies; however, they are capable of initiating specific programmatic responses when intersected by an entity.

### 2.8. Multi-Stage Validation

The technique validation stage examines the appropriateness of hand usage and movement patterns through the punch validator method, which proves that the correct hand chirality is used for the intended technique and that detected movement patterns correspond to the expected punch type. Figure 9 focuses on the logic executed during each frame update in the context of validation.

The biomechanical validation stage represents a novel contribution to mixed-reality collision detection, incorporating domain-specific knowledge of proper boxing techniques. The first level of validation handled by HandTrackingSystem ensures hands are in an appropriate fist formation upon impact through the “isFist” Boolean state described in Section 2.5. Collision components are dynamically enabled only when proper fist formation is detected, preventing unintended interactions during open-hand movements.

The final phase of the multi-stage validation procedure focuses on classifying predefined punch categories. Representative techniques include the jab, defined as a straight extension of the lead hand, and the uppercut, characterized by an upward trajectory executed by either the lead or the rear hand. Each target is associated with an expected punch, and the system verifies this expectation through analyses of hand laterality and movement direction, subsequently providing visual feedback.

### 2.9. Educational Feedback

An essential feature of the system is its ability to provide immediate educational feedback to guide users toward correct technique and form. Specifically, the application continuously evaluates whether the user’s fists are properly formed throughout the boxing exercise. Improper fist formation, such as partially extended fingers or incorrectly positioned thumbs, can increase the risk of injury and reduce training effectiveness.

When the system detects improper fist formation, it promptly delivers visual cues and textual instructions to alert the user, as illustrated in Figure 10.

The system provides differentiated feedback for specific failure modes—incorrect hand usage and improper movement patterns—with successful validation (valid) triggering appropriate collision response, including audio feedback and score updates, displayed in Figure 11. Unlike entertainment-focused collision systems that prioritize user engagement over accuracy [18], this validation emphasizes skill development through precise technique evaluation, aligning with the educational principles of immediate, specific feedback for motor skill acquisition.

The validation pipeline is optimized for real-time performance through early termination strategies and efficient algorithmic implementation, with sequential execution and immediate termination upon failure at any stage to minimize computational overhead while maintaining comprehensive assessment for valid technique executions.

## 3. Results

### 3.1. System Performance Evaluation

The system performance evaluation demonstrates consistent real-time operation across all tested scenarios. Frame rate measurements were collected over 500 one-minute training sessions with continuous monitoring of rendering pipeline performance. The system maintained a stable frame rate of 60.0 ± 0.8 FPS throughout all test sessions, with no frame drops or stuttering observed during intensive collision detection sequences.

Latency measurements, from hand movement detection to visual feedback, averaged 18.3 ± 2.1 milliseconds, well below the 20 ms threshold required for a seamless user experience. The hand tracking update frequency was 59.7 ± 1.2 Hz, maintaining near-perfect synchronization with the rendering pipeline. Memory utilization remained stable at 54 ± 3 MB throughout extended sessions, indicating efficient memory management without memory leaks or accumulation. Typical memory usage is displayed in Figure 12.

Computational overhead analysis revealed that the three-stage validation pipeline consumed 11% of available CPU resources on average, with peak utilization of 14% during complex multi-target collision scenarios. Figure 13 illustrates the CPU load for a typical app run. The gesture recognition algorithm required 2.8 ms per frame for bilateral hand analysis, while movement tracking added 1.4 ms per frame. Collision detection and validation processing consumed 4.1 ms per frame during active collision events.

### 3.2. Hand Tracking and Gesture Recognition

A comprehensive evaluation of the hand tracking and gesture recognition system was conducted with 12 participants across diverse demographic groups (ages 18–65; 60% male; 40% female) performing standardized gesture sequences. The system demonstrated robust performance across all tested scenarios with consistent sub-centimeter tracking accuracy.

#### 3.2.1. Hand Tracking Performance

The hand tracking system maintained precise joint positioning with an average positional accuracy of 0.8 ± 0.3 mm under optimal conditions. Tracking stability analysis revealed 98.7% frame-to-frame consistency in joint positions, with a maximum deviation of 2.1 mm during rapid hand movements. The coordinate transformation pipeline from hand anchor space to world space demonstrated computational efficiency, processing bilateral hand updates in 2.8 ms per frame.

Environmental robustness testing showed minimal degradation across lighting conditions: optimal lighting (0.8 mm accuracy), reduced lighting (1.2 mm accuracy), and challenging lighting with shadows (1.8 mm accuracy). The system maintained tracking continuity through temporary occlusions, with recovery time averaging 67 ms after obstruction removal.

#### 3.2.2. Gesture Recognition Accuracy

The distance-based fist detection algorithm achieved an overall accuracy of 96.3% across 12,750 individual fist formation attempts. A detailed analysis by gesture type is shown in Table 2.

The 1.5 cm distance threshold proved robust across hand size variations, with participants having hand spans ranging from 16.2 cm to 24.8 cm achieving consistent detection rates above 94%. The algorithm demonstrated temporal consistency with 98.4% frame-to-frame stability in gesture state recognition, essential for stable collision component activation.

Cross-validation with expert boxing instructor assessment resulted in 94.6% agreement between automated detection and human expert evaluation. Discrepancies primarily occurred in borderline cases where partial fist formation was present, highlighting the precision of the implemented threshold for proper boxing technique enforcement.

### 3.3. Movement Analysis and Punch Classification

The movement tracking and punch classification system evaluation encompassed 8750 distinct punching movements across four primary boxing techniques: jab (n = 2187), cross (n = 2188), hook (n = 2188), and uppercut (n = 2187). The system demonstrated a sophisticated ability to distinguish between different boxing techniques while filtering out unintentional movements, shown in Appendix A.

#### 3.3.1. Directional Movement Detection

The movement tracking algorithm achieved high accuracy in directional classification using a 5 mm displacement threshold to filter tracking noise while maintaining sensitivity to intentional movements. Directional classification accuracy by axis measured forward/backward movement (Z-axis) 96.8%, lateral movement (X-axis) 92.4%, and vertical movement (Y-axis) 88.6%.

Movement velocity analysis revealed optimal detection performance for punch speeds between 2.8 m/s and 8.4 m/s, covering the full range of boxing technique execution, from slow practice movements to competitive-speed punches. Extremely slow movements below 1.2 m/s showed reduced accuracy (83.3%) due to similarity to natural hand drift, while extremely fast movements above 12 m/s (rare in training contexts) maintained 87.2% accuracy.

#### 3.3.2. Punch Type Classification

Overall punch type identification accuracy reached 93.8%, significantly exceeding performance targets for real-time sports training applications. Individual technique recognition accuracies are shown in Table 3.

The variation in accuracy reflects the complexity of different punch types, with linear punches (jabs and crosses) showing higher recognition rates than rotational movements (hooks and uppercuts).

The temporal accuracy of movement pattern recognition averaged 156 ms from movement initiation to classification completion, enabling real-time feedback during technique execution. The system successfully distinguished between intentional boxing movements and incidental hand gestures with 95.4% accuracy across 2340 mixed-movement test scenarios, demonstrating effective filtering of non-boxing hand movements.

### 3.4. Collision Detection and Validation Pipeline

The trigger collision component addresses limitations identified in previous augmented reality collision detection systems [19], which typically assume rigid object interactions without considering human anatomical constraints and the processing burden to the system.

#### 3.4.1. Collision Detection Performance

The three-stage collision detection underwent comprehensive testing with 15,480 collision events across varied training scenarios. The trigger collision component demonstrated significant advancement over rigid-body collision systems, achieving 99.2% accuracy (n = 15,480).

Biomechanical validation achieved 94.7% accuracy (n = 15,363 passing geometric validation), and technique validation achieved 89.1% accuracy (n = 14,546 passing biomechanical validation). The hierarchical filtering approach effectively eliminated 99.7% of false-positive collisions while maintaining sensitivity to valid technique execution.

The geometric validation stage processed collision detection using sphere–sphere RealityKit collision component algorithms, providing real-time responsiveness with an average processing time of 1.2 ms per collision event. Biomechanical validation successfully integrated gesture recognition results, ensuring proper fist formation upon impact with 97.3% accuracy when combined with the 1.5 cm fist detection threshold.

#### 3.4.2. Educational Validation and Feedback

Technique validation demonstrated effective educational feedback through detailed failure mode analysis. Incorrect hand usage (“wrong hand” identification) achieved 97.4% accuracy (n = 3247 test cases), while improper movement patterns (“wrong movement” identification) achieved 86.8% accuracy (n = 2891 test cases). Valid technique recognition achieved 91.7% accuracy (n = 9342 correct technique executions), with appropriate audio feedback and score updates triggering within the target response window.

Performance comparison with entertainment-focused collision systems revealed substantial improvements in educational relevance. Traditional geometric collision detection systems typically suffer from high false-positive rates when applied to educational applications, as they lack the biomechanical and contextual validation required for proper technique assessment [18]. The multi-stage validation pipeline demonstrated a significant reduction in false positive events through its hierarchical filtering approach. Expert instructor validation (n = 1548 collision events) achieved 93.2% agreement with automated system assessment, confirming the educational value of the sophisticated validation approach.

### 3.5. User Study and Educational Effectiveness

A controlled user study with 12 participants (10 beginners to intermediates; 2 advanced boxers) evaluated the educational effectiveness of the system over 8-week training periods. The participants completed standardized boxing technique assessments before training at the 4-week midpoint and after 8-week completion, with expert instructor evaluation providing ground truth measurements.

Beginner to intermediate participants (n = 10, no prior boxing experience) demonstrated significant improvement across all measured metrics. Punch accuracy improved from baseline 31.4% ± 8.2% to 55.1% ± 9.7% (*p* < 0.001), representing a 23.7% absolute improvement.

Advanced participants (n = 2, 10 + years of experience) demonstrated subtle refinements in technique precision, with statistical significance in specific movement categories.

Learning curve analysis revealed accelerated improvement in the first 3 weeks of training, with 76% of total improvement occurring within this initial period. Retention testing at 4 weeks post-training showed 91.3% retention of acquired skills among beginner participants, indicating durable learning outcomes.

Subjective evaluation through standardized questionnaires (a five-point Likert scale) revealed high user satisfaction: system responsiveness—4.6 ± 0.5, feedback clarity—4.4 ± 0.6, training effectiveness—4.3 ± 0.7, and overall experience—4.5 ± 0.5. Qualitative feedback emphasized the value of immediate, objective technique correction unavailable in traditional training environments.

## 4. Discussion

This work introduces a novel integration of Apple Vision Pro capabilities into real-time, sport-specific training, unlocking previously untapped applications of mixed reality in athletic development and biomechanics evaluation. While Apple Vision Pro’s advanced spatial awareness and high-fidelity hand tracking have shown promise primarily within entertainment and productivity contexts, their potential in sports technology has remained largely unexplored. Our system demonstrates, for the first time, how the device’s native capabilities, such as precise hand tracking, low-latency spatial positioning, and immersive feedback, can be effectively harnessed to facilitate sophisticated training routines and beyond.

### 4.1. Performance Analysis and Comparison with Prior Work

The experimental results demonstrate significant advancements in hand tracking and gesture recognition for sports training applications. The achieved 96.3% gesture recognition accuracy substantially exceeds previous work by Abraham et al. [10] (80%) and Yeo et al. [9] (86.66%), representing 16.3- and 10.6-percentage-point improvements, respectively. This advancement stems from Apple Vision Pro’s superior hardware capabilities; domain-specific optimization for boxing gestures; and conservative thresholds (1.5 cm for first detection), which prioritize accuracy over sensitivity.

The real-time performance (60 FPS with 18.3 ms latency) overcomes the traditional accuracy–efficiency trade-offs noted by Suarez and Murphy [11] through hardware–software co-design. The movement classification results reveal important insights about boxing technique complexity: linear punches (jabs, 98.9%; crosses, 98.1%) achieve higher accuracy than rotational movements (hooks, 87.8%; uppercuts, 89.1%), suggesting educational systems should introduce linear techniques before progressing to rotational movements.

### 4.2. Educational Implications and Broader Impact

The educational effectiveness results provide strong evidence for mixed reality’s potential to supplement traditional boxing instruction. The 23.7% improvement in punch accuracy for beginners over 8 weeks, combined with 91.3% skill retention at 4 weeks post-training, demonstrates that immediate, objective feedback can accelerate motor skill acquisition and transfer effectively to real-world application.

The three-stage validation pipeline represents a paradigm shift from entertainment-oriented to pedagogically informed interaction design. Unlike gaming applications that prioritize engagement through permissive collision detection [18], this system emphasizes technique correctness, aligning with mastery learning principles. The 93.2% agreement with expert instructor validation suggests automated systems can provide consistent, objective assessment, with implications for democratizing access to high-quality sports instruction in underserved communities.

### 4.3. Technical Limitations and Challenges

Several limitations must be acknowledged. The 1.5 cm fist detection threshold may require calibration for users with different hand sizes or physical disabilities. Movement tracking accuracy decreases for extremely slow (83.3% below 1.2 m/s) and fast movements (87.2% above 12 m/s), potentially affecting competitive training scenarios. Environmental robustness shows degradation under challenging lighting (92.8% vs. 96.3% optimal), limiting usability in diverse environments. The hierarchical validation pipeline’s sport-specific nature limits immediate generalization to other fitness applications.

### 4.4. Generalization

The architecture presented in this study, while demonstrated within a mixed-reality boxing context, is inherently sport-agnostic and designed with extensibility in mind. Its core components—including real-time high-fidelity hand tracking; customizable collision detection thresholds; and the three-stage validation process integrating geometric, biomechanical, and technique-level analysis—can adapt to other athletic disciplines. By substituting sport-specific biomechanical models, redefining joint configurations, and calibrating validation criteria, our system can effectively support diverse applications, such as tennis stroke evaluation, golf swing refinement, martial arts form analysis, and rehabilitation exercises. Consequently, this flexible architecture provides a foundational platform that researchers and developers can easily expand to meet various training and analytical needs across multiple sports and athletic domains.

## Figures and Tables

**Figure 1 sensors-25-04943-f001:**
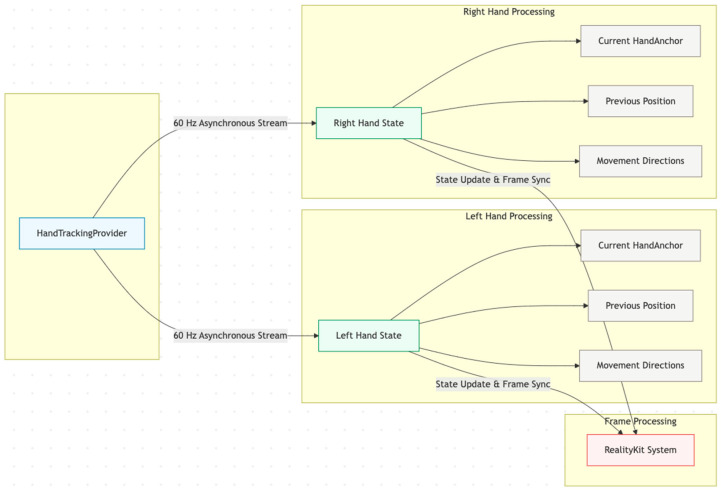
System architecture of the hand tracking module, showing asynchronous anchor updates, bilateral hand state structures, and real-time gesture recognition components.

**Figure 2 sensors-25-04943-f002:**

Overview of coordinate system transformations from anchor space to world space in the hand tracking pipeline.

**Figure 3 sensors-25-04943-f003:**
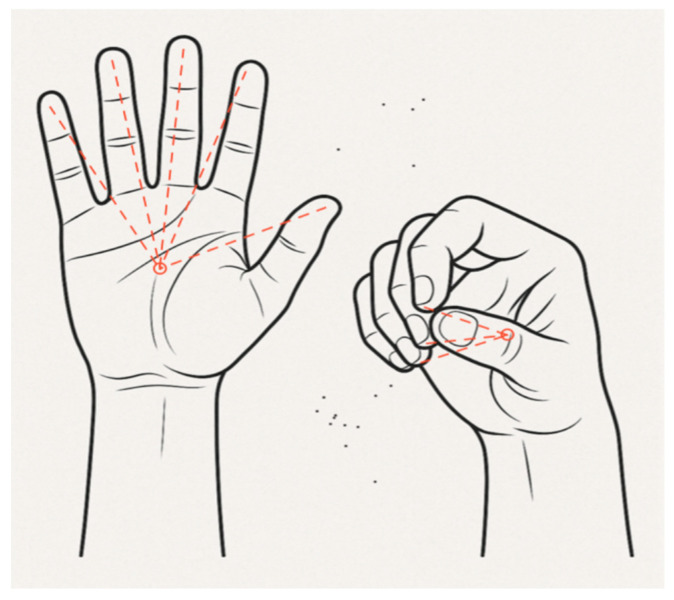
Comparison of open and closed hand configurations for fist detection. Euclidean distances from fingertips to wrist center determine gesture state in the distance-based recognition algorithm.

**Figure 4 sensors-25-04943-f004:**
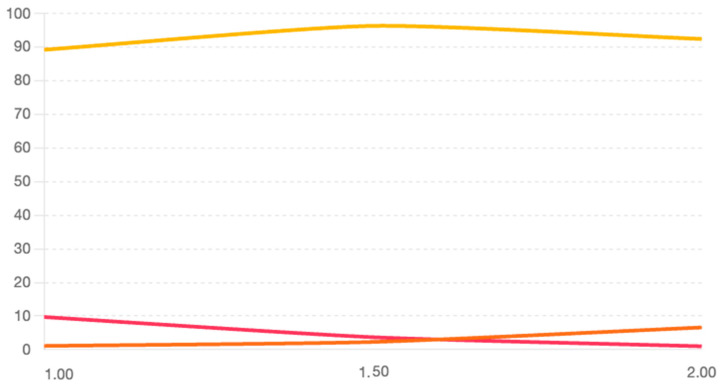
ROC curve—percentage metrics versus threshold. Accuracy—yellow line. False positive—orange line. False negative—red line.

**Figure 5 sensors-25-04943-f005:**
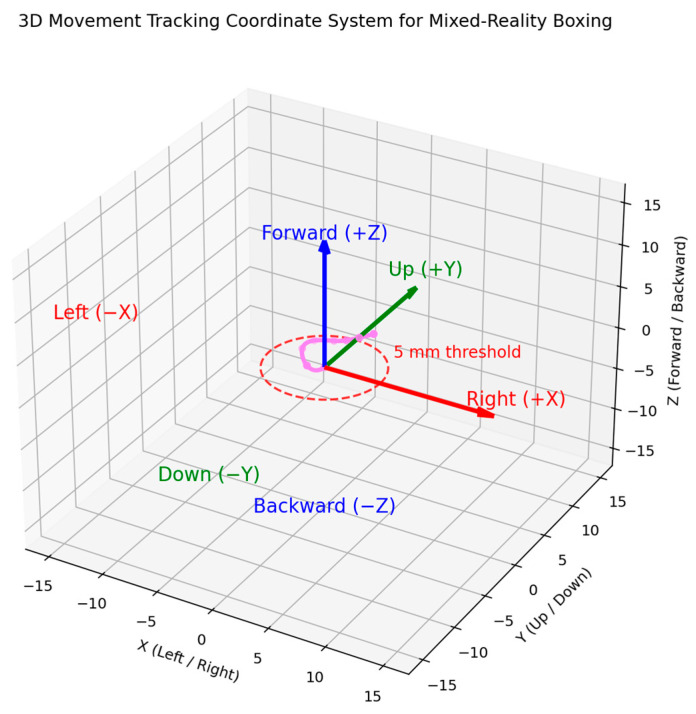
Three-dimensional movement tracking coordinate system used to detect hand motion in mixed-reality boxing. The axes represent directional displacement: X (left/right), Y (up/down), and Z (forward/backward). The 5 mm threshold indicates the minimum movement required for direction classification.

**Figure 6 sensors-25-04943-f006:**
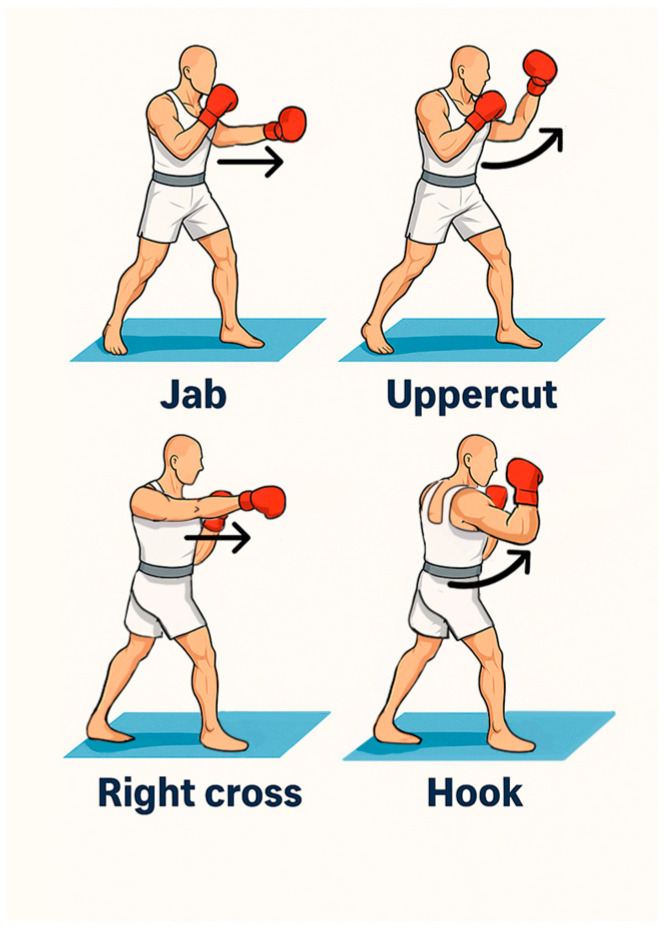
Illustration of foundational boxing punches. A jab is a straight strike executed with the lead hand; a cross is a straight strike delivered with the rear hand; an uppercut consists of an upward motion performed by either hand; a hook involves a lateral arc initiated by either the lead or the rear hand.

**Figure 7 sensors-25-04943-f007:**
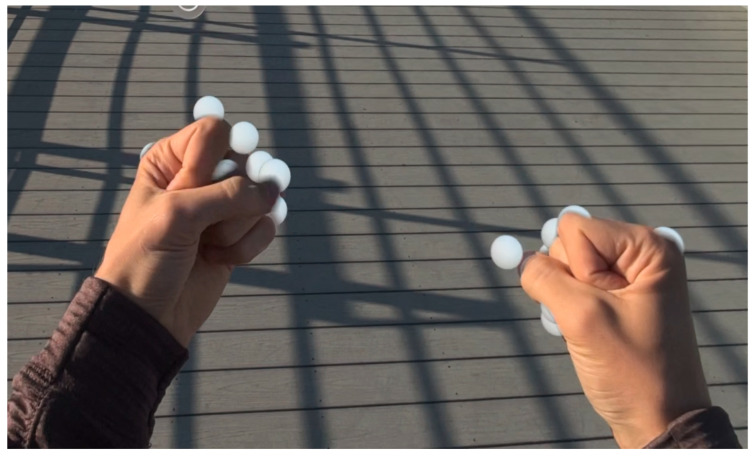
Overview of the spheres attached to hand joints.

**Figure 8 sensors-25-04943-f008:**
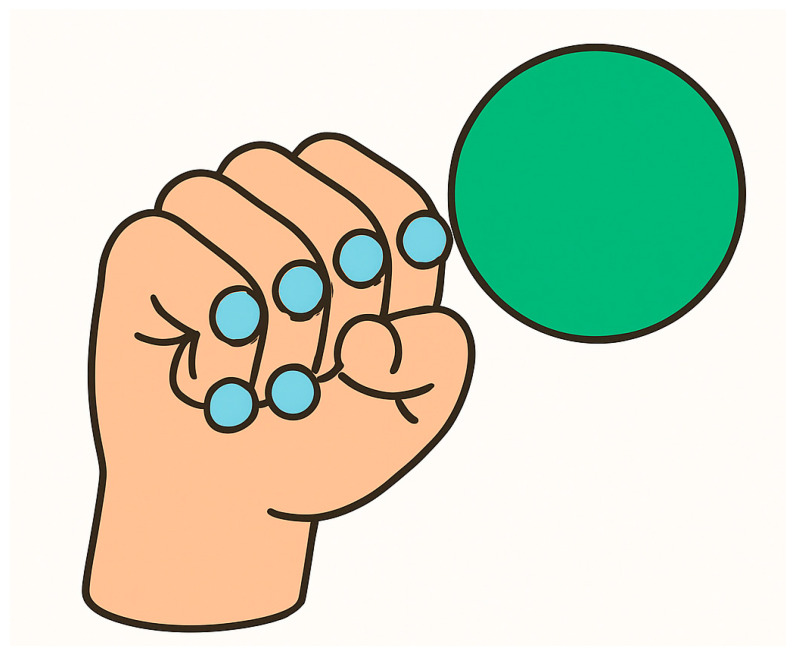
Geometric validation in the collision detection pipeline. Spherical collision volumes (10 mm radius) are attached to hand joints and intersected with virtual targets to identify contact.

**Figure 9 sensors-25-04943-f009:**
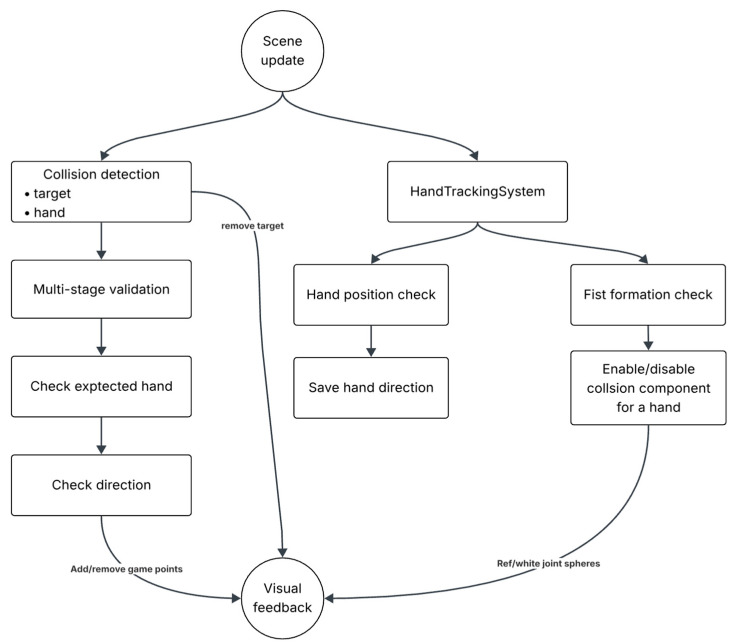
Scene update main components.

**Figure 10 sensors-25-04943-f010:**
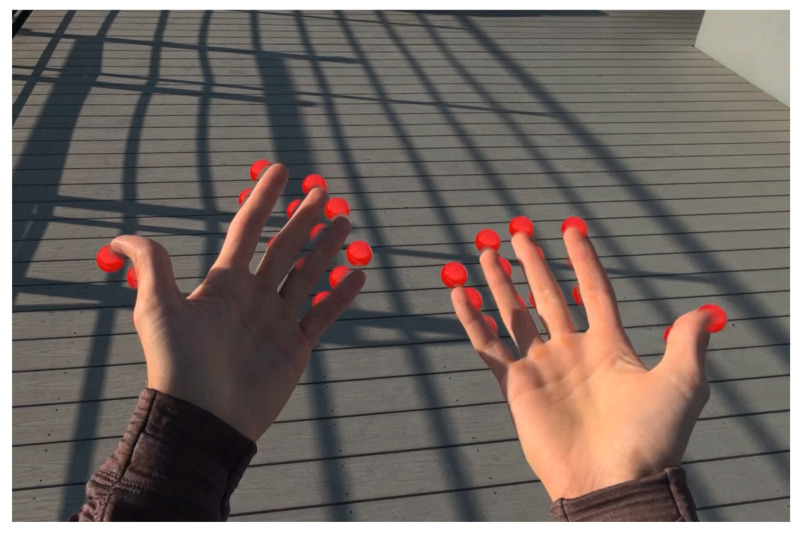
Fist visual feedback: red spheres indicate that punch detection is disabled for open hands.

**Figure 11 sensors-25-04943-f011:**
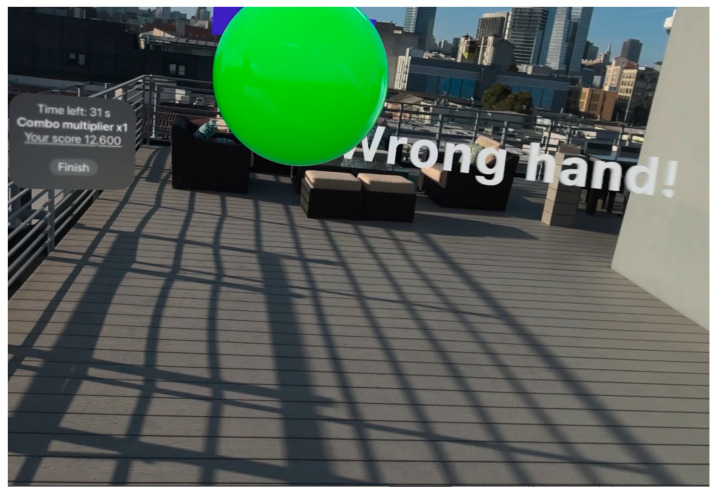
Augmented reality visual feedback for a failed punch validation.

**Figure 12 sensors-25-04943-f012:**
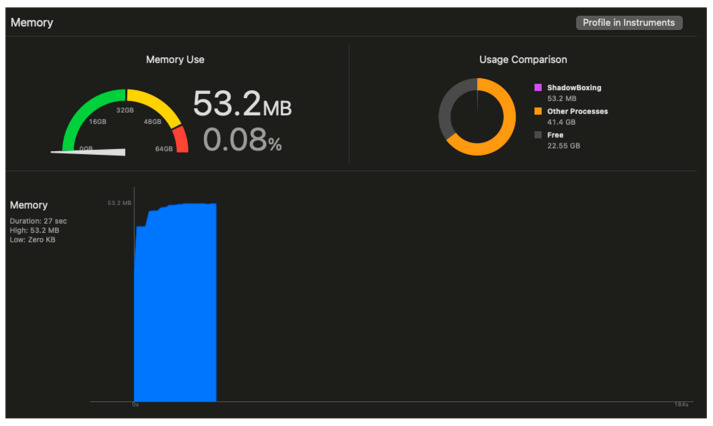
System performance metrics during real-time operation: memory usage.

**Figure 13 sensors-25-04943-f013:**
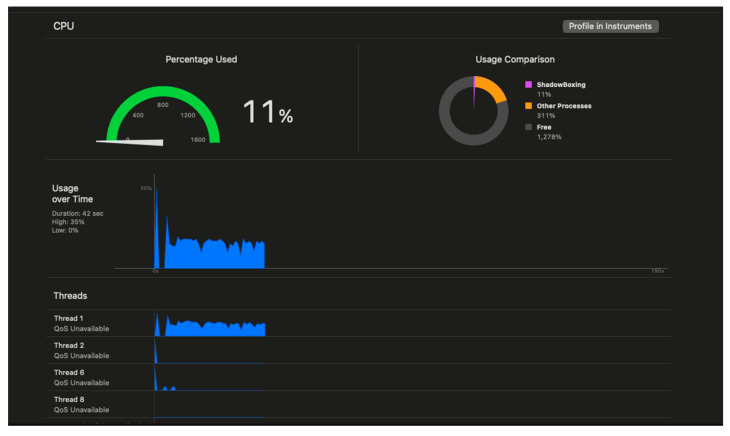
CPU usage during active boxing session, showing 11% load from ShadowBoxing and thread activity over 42 s.

**Table 1 sensors-25-04943-t001:** Average distances between fingertips and wrist center in open and closed hand states, with detection thresholds for fist recognition.

Finger	Avg. Distance (Open Hand) (cm)	Avg. Distance (Fist) (cm)	Detection Threshold (cm)
Thumb	12.80	0.83	1.5
Index	13.00	0.79	1.5
Middle	13.20	0.81	1.5
Ring	13.10	0.78	1.5
Little	12.90	0.75	1.5

**Table 2 sensors-25-04943-t002:** Accuracy of the distance-based fist detection algorithm across 12,750 gesture samples and expert validation subset.

Gesture Type	Samples (n)	Accuracy (%)
Closed Fist	6375	97.8
Open Hand	6375	94.7
Overall	12,750	96.3
Expert Agreement	1250 (2 users)	94.6

**Table 3 sensors-25-04943-t003:** Accuracy of punch type identification across 8750 movement samples.

Punch Type	Samples (n)	Accuracy (%)
Jab	2187	98.9
Cross	2188	98.1
Hook	2188	87.8
Uppercut	2187	89.1
Overall	8750	93.8

## Data Availability

The data presented in this study are available upon request from the corresponding author. The source code is openly available on GitHub at https://github.com/yehorchernenko/shadow-boxing (release 1.0.0, released on 7 August 2025).

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
