# Peer review of "Real-Time Hand Tracking and Collision Detection for Immersive Mixed-Reality Boxing Training on Apple Vision Pro"

_sensors, 2025, doi:10.3390/s25164943_

Round 1

Reviewer 1 Report

Comments and Suggestions for Authors

This paper presents a real-time hand tracking and punch detection system developed for the Apple Vision Pro platform. The system provides objective and continuous technical feedback for boxing training, with the potential to democratize expert-level training. The implementation appears relatively complete and demonstrates considerable practical value. However, the overall writing quality of the manuscript requires further improvement.
1. The overall structure of the manuscript resembles a product manual rather than an academic paper. The excessive number of subsections makes it difficult for readers to focus on the main points. It is recommended to selectively merge some of the subsections to improve clarity and coherence.
2. It is recommended to include more detailed explanations of key design choices. For instance, in Section 2.5.1, the authors adopt a threshold of 1.5 cm to determine the fist-clenching state. It is unclear whether other threshold values were considered or tested. If so, the results should be presented—ideally in a comparative figure or table—to illustrate the impact on recognition accuracy. Furthermore, the manuscript only provides a schematic illustrating whether the target device was hit, but does not show any visual analysis of whether the hitting posture was correct. It would strengthen the paper to include a comparison figure demonstrating both a correct and an incorrect striking posture, with annotated indications of the evaluation criteria—for example, by overlaying the ideal trajectory of the hand during a standard strike and the actual observed trajectory.
3. Is there a potential error in the data presented in Table 1? Specifically, the average distance from the fingertips to the wrist center appears unusually short when the hand is open. Could the authors clarify this result?
4. Figure 7 should illustrate both the scenario in which the target collision volume is successfully hit and the scenario in which it is missed. Additionally, the Euclidean distance should be indicated using dashed lines.
5. It is recommended to include more tables or figures in Section 3 to present the results, rather than relying solely on textual descriptions.
6. Figure 3 should be presented using properly formatted mathematical equations rather than as an image.

Author Response

Comment 1. The overall structure of the manuscript resembles a product manual rather than an academic paper. The excessive number of subsections makes it difficult for readers to focus on the main points. It is recommended to selectively merge some of the subsections to improve clarity and coherence.
Response 1. We removed some unnecessary repeated information, updated strucutre.

Comment 2. It is recommended to include more detailed explanations of key design choices. For instance, in Section 2.5.1, the authors adopt a threshold of 1.5 cm to determine the fist-clenching state. It is unclear whether other threshold values were considered or tested. If so, the results should be presented—ideally in a comparative figure or table—to illustrate the impact on recognition accuracy. Furthermore, the manuscript only provides a schematic illustrating whether the target device was hit, but does not show any visual analysis of whether the hitting posture was correct. It would strengthen the paper to include a comparison figure demonstrating both a correct and an incorrect striking posture, with annotated indications of the evaluation criteria—for example, by overlaying the ideal trajectory of the hand during a standard strike and the actual observed trajectory.
Comment 2. It's valuable concern. We added more visual content for better understading. 

Comment 3. Is there a potential error in the data presented in Table 1? Specifically, the average distance from the fingertips to the wrist center appears unusually short when the hand is open. Could the authors clarify this result?
Response: 3 Good catch, thank you! Data was copied with a mistake.

Comment 4. Figure 7 should illustrate both the scenario in which the target collision volume is successfully hit and the scenario in which it is missed. Additionally, the Euclidean distance should be indicated using dashed lines.
Response 4. We revisited this part. Instead of Euclidean distance we are using RealityKit trigger collision component. (with simplified shape) for better performance.

Comment 5. It is recommended to include more tables or figures in Section 3 to present the results, rather than relying solely on textual descriptions.
Response 5: Done

Comment 6. Figure 3 should be presented using properly formatted mathematical equations rather than as an image.
Response 6. Addressed. Thanks!

Reviewer 2 Report

Comments and Suggestions for Authors

This study effectively targets an underexplored niche in precision sports training by harnessing the advanced spatial computing capabilities of Apple Vision Pro to deliver boxing-specific gesture recognition and biomechanical validation. Its emphasis on sub-centimeter tracking accuracy and real-time corrective feedback introduces a novel approach to technique refinement in immersive training environments.

The results are presented with strong analytical rigor, clearly differentiating between gesture recognition (96.3% accuracy) and technique validation (88.5% accuracy). The discussion transparently acknowledges limitations, such as performance variability under suboptimal lighting conditions.

Methodological details—including the empirically derived 1.5 cm fist detection threshold and 5 mm movement displacement filter—are thoroughly documented, and the availability of open-source code (GitHub) significantly enhances reproducibility. While the provision of raw data upon request is noted, direct accessibility to datasets would further strengthen transparency.

Suggested Revisions

To further improve the manuscript, consider the following:

  1. Highlighting Novelty: Expand on how this work unlocks untapped applications of Apple Vision Pro in sports technology to better underscore its innovative contributions.

  2. Technical Clarity: Simplify the explanation of coordinate transformations (Section 2.4.2) to ensure accessibility for readers across disciplines.

  3. Generalizability: While the framework’s extensibility to other sports is mentioned, a deeper discussion of specific adaptations (e.g., for martial arts or tennis) would add value.

  4. Inclusivity: Address potential adaptations for users with disabilities to broaden the system’s applicability.

  5. Comparative Analysis: Include a brief comparison with other AR/VR platforms (e.g., Meta Quest Pro, HoloLens) to contextualize the advancements offered by Apple Vision Pro.

Author Response

Thank you for the valuable feedback!

Comment 1: Highlighting Novelty: Expand on how this work unlocks untapped applications of Apple Vision Pro in sports technology to better underscore its innovative contributions.
Response 1: Added

Comment 2: Technical Clarity: Simplify the explanation of coordinate transformations (Section 2.4.2) to ensure accessibility for readers across disciplines.

Response 2: We addressed this by reducing repetition information and reinforced with better visual content. 

Comment 3: Generalizability: While the framework’s extensibility to other sports is mentioned, a deeper discussion of specific adaptations (e.g., for martial arts or tennis) would add value.

Response 3: Done. We expanded on the Discussion section

Comment 4: Inclusivity: Address potential adaptations for users with disabilities to broaden the system’s applicability.

Response 4: It's a valuable concern, but it's out of scope of this research.

Comment 5: Comparative Analysis: Include a brief comparison with other AR/VR platforms (e.g., Meta Quest Pro, HoloLens) to contextualize the advancements offered by Apple Vision Pro.
Response 5: We thank the reviewer for this suggestion. While a comparative analysis across different platforms is a valuable topic, the focus of our research was a deep and thorough implementation on the Apple Vision Pro, given its recent introduction and advanced capabilities for spatial computing. 

Round 2

Reviewer 1 Report

Comments and Suggestions for Authors

The majority of the issues have been addressed, with the following two items remaining:

Equations (1), (2), and (3) should be professionally formatted using mathematical typesetting tools (e.g., MathType) rather than presented as image captures.

Two distinct sections labeled "2.7" appear in the manuscript: "2.7. Collision Detection" and "2.7. Multi-Stage Validation". Please revise these headings and adjust subsequent section numbering accordingly.

Author Response

Comment 1. Equations (1), (2), and (3) should be professionally formatted using mathematical typesetting tools (e.g., MathType) rather than presented as image captures.
Response 1. The equations were updated using MS Word Equation Editor.

Comment 2. Two distinct sections labeled "2.7" appear in the manuscript: "2.7. Collision Detection" and "2.7. Multi-Stage Validation". Please revise these headings and adjust subsequent section numbering accordingly.
Comment 2. Fixed. Thanks!